# Seasonal Variation and Geographical Distribution of COVID-19 across Nigeria (March 2020–July 2021)

**DOI:** 10.3390/vaccines11020298

**Published:** 2023-01-29

**Authors:** Jude Eguolo Moroh, David Chinaecherem Innocent, Uchechukwu Madukaku Chukwuocha, Advait Vasavada, Ramesh Kumar, Mohammad Arham Siddiq, Mohammed Amir Rais, Ali A. Rabaan, Wafa M. Alshehri, Areej M. Alharbi, Mohammed A. Binateeq, Muhammad A. Halwani, Tareq Al-Ahdal, Bijaya Kumar Padhi, Ranjit Sah

**Affiliations:** 1Department of Public Health, Federal University of Technology Owerri, Owerri 460114, Nigeria; 2MP Shah Medical College, Jamnagar 361008, India; 3Health Services Academy, Islamabad 44000, Pakistan; 4Department of Medicine, Jinnah Sindh Medical University, Karachi 75510, Pakistan; 5Faculty of Medicine of Algiers, University of Algiers, Alger Ctre 16000, Algeria; 6Molecular Diagnostic Laboratory, Johns Hopkins Aramco Healthcare, Dhahran 31311, Saudi Arabia; 7College of Medicine, Alfaisal University, Riyad 11533, Saudi Arabia; 8Department of Public Health and Nutrition, The University of Haripur, Haripur 22610, Pakistan; 9Riyadh Regional Laboratory, Molecular Microbiology and Cytogenetics Department, Riyadh 11425, Saudi Arabia; 10Department of Medical Microbiology, Faculty of Medicine, AlBaha University, AlBaha 65528, Saudi Arabia; 11Institute of Global Health, Heidelberg University, Neuenheimerfeld130/3, 69120 Heidelberg, Germany; 12Department of Community Medicine, School of Public Health, Postgraduate Institute of Medical Education and Research, Chandigarh 160012, India; 13Tribhuvan University Teaching Hospital, Institute of Medicine, Kathmandu 44600, Nepal; 14Dr. D.Y. Patil Medical College, Hospital and Research Centre, Dr. D. Y. Patil Vidyapeeth, Pune 411018, Maharashtra, India

**Keywords:** COVID-19, Nigeria, pandemic, outbreak

## Abstract

Globally, the novel corona virus infection has continued to witness a growing number of cases since December 2019 when the outbreak was discovered and noted in China. Despite this has not been well studied for the case of COVID-19, human contact, public moveableness and environmental variables could have an impact onairborne’spropagation and virus continuance, such as influenza virus. This study aimed to determine the seasonal variation and geographical distribution of COVID-19 across Nigeria. An internet based archival research design was employed for this study on the seasonal variation and geographical distribution of COVID-19 across Nigeria. This involved the use of goggle mobility data and world map on Corona Virus Infection (COVID-19). The search strategy for getting information for this research was done electronically. The keywords in the case search using the goggle mobility software was “COVID-19 Update”, “COVID-19 Update in Nigeria”, ‘COVID-19 Winter Report’, “COVID-19 Case Fatality March 2020–July 2021”, “COVID-19 Case Fatality in Nigeria”. The data gotten from the goggle motor updates were entered into Statistical Package for the Social Sciences (SPSS) which was used in the analysis of the study. Results from the study, reported that official COVID-19 cases number was significantly higher in the Dry season (October 2020–April 2021) with 59.0% (127,213) compared to 41.0% (85,176) in the wet/rainy season (May–September) it revealed that the dry and rainy seasons had a COVID-19 prevalence of 0.063 and 0.041 respectively. Further results from the study showed that the prevalence of COVID-19 was 0.07% in the North-Central, 0.04% in both the North-East and North-West, 0.03% in the South-West, 0.09% in the South-South, and the highest prevalence of 0.16% in the South-East. Considering the case Fatality rate of COVID-19 during the Dry and Wet Seasons. The study revealed that North-Central had a death toll of 196 (10.4%) out of 9457 confirmed COVID-19 cases hence a fatality of 2.07. Fatality rate of 1.49% in South western Nigeria, South-South Nigeria, 1.49%, South-East accounted to a fatality rate of 1.25%. Nigeria based on the finding of this study records increased fatality in Dry season over wet seasons. The study concluded that prevalence of COVID-19 varies in seasons in Nigeria Hence; further Data and Meteorological analysis on weather variations towards the SARS-CoV-2 Virus spread should be evaluated by future researchers. It is imperative to ensure strict and controlled application of social measures, such as social distancing, mandatory wearing of non-medical masks to prevent droplets from entering the respiratory tract, screening of affected patients along with quarantine is essential to defeat and improve infection control.

## 1. Introduction

The novel coronavirus infection has continued to witness a growing number of cases since December 2019, when the outbreak was discovered and noted in China [1,2]. In 2020 World Health Data, it was established that COVID-19 clearly has stronger spread rates among people [1]. It was reported that the official number of registered cases was 17,396, and 943, and the mortality numbers were 675,060, provided through August 2020 worldwide, leading to the decision to impose total containment worldwide after a global pandemic was officially declared [1,3,4]. According to Oliveiros et al [5] in addition to human contact, public mobility and environmental variables, such as the influenza virus, could have an impact on airborne virus spread and virus vitality, though this was not well explained for COVID-19.

The geographic distribution of COVID-19, as illustrated in a report established that the coronavirus 2 (SARS-CoV-2) is different hereditarily from both the coronavirus 1 and the Middle East respiratory syndrome coronavirus (MERS-CoV), and thus, it can develop in a different way depending on climatic variables [6]. Gradually, we have seen a trend of augmented numbers of patients, not only in the source country of China but as well in Europe and America. In addition to the transmission of COVID-19 from animals to humans, inter-human contagiousness has also been demonstrated [2]. As a result of the rapid spread of COVID-19 in the world and the emergence of a pandemic, Italy is one of the other European nations where the propagation of the infection has been reported as horribly worrying. The number of cases in the country had reached 80,589 by 26 March 2020. In addition, European countries such as France, Belgium, France, Austria, Switzerland, and the United Kingdom have also reported large numbers of COVID-19 [1]. And as the first declared case in Latin America was a Brazilian citizen with recent travel history to Italy, all this evidence and statistics report the importance of human-to-human transmission of COVID-19 [1]. Some publications have asserted the epidemiological hypothesis that dry, cold environments with lower absolute humidity and temperature favor the survival and transmission of viral infections via droplets, whereas humid, warm environments with higher absolute humidity and temperature suppress such transmission [7,8,9]. A study reported that humidity, contents of the atmospheric water, and temperature are highly correlated with other viral spreads such as influenza [10]. The influenza virus’s lifespan, whether airborne or on surfaces, is clearly longer in dry, cold air, which increases its ability to be contagious [7,11]. Despite that, there is little scientific evidence on the impact of both temperature and humidity on the new Corona virus’s propagation.

Another publication declared the new COVID infection a pandemic with sudden transmission and brutally overwhelming health services due to cases requiring intensive management all over the world [12]. It is imperative to note that there is a rational debate in the adaptation plans for COVID-19 about whether the virus would spread more slowly in humid and warm weather. Conversely, in a publication by Bogochet et al. [13] as reported, it is found that, up to 1 August 2020, 216 countries and regions around the world had shown local spread of COVID-19, including all climatic zones, from dry and cold to wet and hot. However, some Asian countries with local spread were popular travel destinations for Chinese travelers, resulting in imported cases [2]. Similarly, countries in Africa and South America with moderately high temperatures showed local spread. Several studies have found that humidity and temperature affect the pattern of global spread of the COVID-19 pandemic [5,14,15,16].

In addition, a 2021 model explained the relation between sensitive airborne particles and relative humidity and temperature and their combination with COVID-19 [6]. It was assumed that COVID-19 was spread exclusively through the inhalation of already infected respiratory droplets. The time it takes for droplets to vaporize and dry, an indicator of the infection rate constant, depends on relative humidity and temperature indoors and outdoors, and the number of infected cases is directly associated with this infection rate constant. The study suggests that relative humidity and temperature affected the spread of COVID-19 [6]. They found that with comparable relative humidity at two sites, COVID-19 spread faster at cooler temperatures than at warmer temperatures, with a negligible effect when including social enforcement [6]. In a model by Araujo and Naimi [16], it is illuminated that respiratory droplets are sensitive to relative humidity and temperature and are associated with COVID-19.

According to a study by Qi et al. [17] conducted in China examining the relations between the daily average relative humidity and temperature and the daily counted cases of COVID-19 in 30 districts using the generalised additive model (GAM), the study revealed that COVID-19 daily case counts were adversely associated with both the average relative humidity and temperature. The study suggested that both daily relative humidity and temperature affected the incidence of COVID-19 in some districts of China [17]. With the increasing prevalence of COVID-19 spread statistics and the extending time between detected cases, it is imperative that studies consider meteorological factors such as humidity, temperature, precipitation, and wind velocity when studying viral transmissions [4]. Altamimi and Ahmed [18] discovered in a model that there is an inverse correlation between doubling time and temperature and humidity, indicating a lower rate of COVID-19 prevalence in the north during the fall and summer. Nevertheless, a research study studied the temperature as a crucial element in the COVID-19-caused infection, where researchers obtained data on both meteorological factors and official cases daily detected in 122 cities in China [19].

Bhattacharjee [20] highlighted to determine whether the coronavirus is associated with seasonality and the probability of assessed transmission. According to the study, the predominant areas of COVID-19 had lower absolute moisture and mean temperature than those where virus propagation was lower [20]. As a result, the study discovered that the COVID-19 specimen obtained through commanded measurements of temperature, humidity, and latitude was consistent with a seasonal behaviour of the respiratory virus, implying that it is a model for the prediction of the possibility of an expressive spread of coronavirus in these regions [20]. The global spread of COVID-19 implies that seasonal differentiation cannot be considered solely as a significant modifier of spread; however, a warmer climate could moderately limit the spread of COVID-19. No evidence indicated that a warmer climate would limit the potency of COVID-19’s spread, making additional actions to restrict transmission less necessary. As a result, it is critical to explain these evidences in light of the coronavirus’s continued global spread.

Regrettably, worldwide, it is seen that the novel coronavirus has altered and overwhelmed the entire economy of nations [19,20,21,22,23]. During their trials, public health authorities have made available different approaches to prevent and/or limit the spread of the virus. Understanding the mechanism of environmental influence on the current pandemic disaster will help to facilitate the right decisions to halt the spread, particularly in warmer and wetter regions where spread rates may have been underestimated. In a developing country like Nigeria, the effectiveness of containment measures depends in part on whether the spread of infection decreases in response to changes in humidity and temperature between seasons.

A report noted that other catastrophic implications of COVID-19 include critical complications like respiratory failure, shock, or multi-organ dysfunction, etc. [24]. Altamimi and Ahmed [18] used the generalised additive log-linear programme to study the effects of relative humidity and temperature on daily reported new coronavirus infections and mortality rate in a Sub-Saharan country like Nigeria. The project revealed that both relative humidity and temperature were negatively associated with new infection rates and mortality reported daily (decreased mortality and infections with higher relative humidity and temperature), suggesting that virus transmission could be alleviated with higher humidity and temperature [18]. In Nigeria, located and bordering the Atlantic Ocean, it is imperative to understand how changes in season influence the outcome of the dreaded pandemic; thus, this study would examine how the variations in seasons could have an impact on the spread of the coronavirus in several sites in Nigeria, reporting that some observed patterns of spread can confirm or deny the conventional idea that the higher the absolute humidity and temperature, the lower the contagiousness and life span of COVID-19.

## 2. Methods

### 2.1. Study Design

An internet based archival research design was employed on the course of this study on the seasonal variation and geographical distribution of COVID-19 across Nigeria. 

### 2.2. Area of Study

Nigeria is located in western Africa on the Gulf of Guinea and has a total area of 923,768 km^2^ (356,669 sq mi), making it the world’s 32nd-largest country. Its borders span 4047 kilometres (2515 mi), and it shares borders with Benin (773 km or 480 mi), Niger (1497 km or 930 mi), Chad (87 km or 54 mi), and Cameroon (including the separatist Ambazonia) 1690 km or 1050 mi. Its coastlineis at least 853 km (530 mi). Nigeria lies between latitudes 4 and 14 N, and longitudes 2 and 15 E. The highest point in Nigeria is Chappal Waddi at 2419 m (7936 ft). The main rivers are the Niger and the Benue, which converge and empty into the Niger Delta. This is one of the world’s largest river deltas, and the location of a large area of Central African mangroves. Nigeria’s most expansive topographical region is that of the valleys of the Niger and Benue river valleys (which merge and form a Y-shape). To the southwest of the Niger is "rugged" highland. To the southeast of the Benue are hills and mountains, which form the Mambilla Plateau, the highest plateau in Nigeria. This plateau extends through the border with Cameroon, where the montane land is part of the Bamenda Highlands of Cameroon. The six geopolitical zones in Nigeria and the states that make up each of the geopolitical zones

(1)North Central (loosely known as Middle Belt):
BenueKogiKwaraNasarawaNigerPlateauFederal Capital Territory
(2)North East:
AdamawaBauchiBornoGombeTarabaYobe
(3)North West:
JigawaKadunaKanoKatsinaKebbiSokotoZamfara
(4)South East:
AbiaAnambraEbonyiEnuguImo
(5)South South (also known as Niger Delta region)
Akwa IbomBayelsaCross RiverRiversDeltaEdo
(6)South West:
EkitiLagosOgunOndoOsunOyo


#### 2.2.1. Climate (Dry and Wet Season)

The rainy season in the country lasts from March to September, and only in the south it is briefly interrupted in August, and the dry season falls on the remaining months of the year (in the north it lasts longer than in other regions) Nigeria has a varied landscape. The far south is defined by its tropical rainforest climate, where annual rainfall is 60 to 80 inches (1500 to 2000 mm) per year. In the southeast stands the Obudu Plateau. Coastal plains are found in both the southwest and the southeast. Mangrove swamps are found along the coast. 

#### 2.2.2. Climate Map of Nigeria

The area near the border with Cameroon close to the coast is rich rainforest and part of the Cross-Sanaga-Bioko coastal forests eco region, an important centre for biodiversity. It is habitat for the drill primate, which is found in the wild only in this area and across the border in Cameroon. The areas surrounding Calabar, Cross River State, also in this forest, are believed to contain the world’s largest diversity of butterflies. The area of southern Nigeria between the Niger and the Cross Rivers has lost most of its forest because of development and harvesting by increased population, with it being replaced by grassland.

Everything in between the far south and the far north is savannah (insignificant tree cover, with grasses and flowers located between trees). Rainfall is more limited to between 500 and 1500 millimeters (20 and 60 in) per year. The savannah zone’s three categories are Guinean forest-savanna mosaic, Sudan savannah, and Sahel savannah. Guinean forest-savanna mosaic is plains of tall grass interrupted by trees. Sudan savannah is similar but with shorter grasses and shorter trees. Sahel savannah consists of patches of grass and sand, found in the northeast. In the Sahel region, rain is less than 500 millimeters (20 in) per year, and the Sahara Desert is encroaching. In the dry northeast corner of the country lies Lake Chad, which Nigeria shares with Niger, Chad and Cameroon (Figure 1)

### 2.3. Study Population

This study on the seasonal variation and geographical distribution of COVID-19 across Nigeria involved an internet review of mobile records of COVID-19 in Nigeria from March 2020–July 2021. This study involved a case search for COVID-19 infection among the geographic distribution.

### 2.4. Sampling

There was no sample size calculation for the study given the nature and scope of the study on the seasonal variation and geographical distribution of COVID-19 across Nigeria. The study compared COVID-19 among several geopolitical regions in Nigeria the seasonal distribution of the infection using the goggle mobility and other related internet search.

### 2.5. Instrument and Method Data Collection

The instrument for data collection was a Goggle Mobility software aimed to obtain relevant information on the seasonal variation and geographical distribution of COVID-19 across Nigeria. This was targeted at getting relevant information on the seasonal variation (winter to summer) and distribution of incidence cases of COVID-19 across Nigeria, targeted to investigate the seasonal variation and distribution in transmission of COVID-19 infection across Nigeria. And to compare between the dry season and wet season plunge in the Case fatality of COVID-19 infection across Nigeria

### 2.6. Search Strategy

This study reviewed information obtained from goggle-motor updates. The search strategy for getting information for this research was done electronically.

#### Keywords in Search Strategy 

The keywords in the case search using the goggle mobility software was “COVID-19 Update”, “COVID-19 Update in Nigeria”, ‘COVID-19 Winter Report’, “COVID-19 Case Fatality March 2020–July 2021”, “COVID-19 Case Fatality in Nigeria”, “COVID-19 Transmission in Nigeria”.

### 2.7. Instrument of Data Collection

The instrument of data collection for this study was a COVID-19 checklist used to obtain information relevant to the study objectives. Records regarding to the COVID-19 in Nigeria was assessed from several websites and recorded on the checklist.

### 2.8. Method of Data Analysis

The data gotten from the goggle motor updates were entered into Statistical Package for the Social Sciences (SPSS) version 23.0 which was used in the analysis of the data gotten from the study into percentages, frequencies, tables and charts (Descriptive Statistics). Also a non parametric test was not used considering the nature of the data obtained for the study. 

### 2.9. Ethical Consideration

A letter of introduction and ethical clearance was obtained from the School of post graduate studies Department of Public Health Ethical clearance committee in Federal University of Technology Owerri (FUTO) before the research was conducted.

## 3. Results 

### 3.1. Prevalence of COVID-19 during the Dry and Wet Seasons across Nigeria (October 2020–April 2021)

Results from Table 1 below, showed that the number of confirmed COVID-19 cases was significantly higher in the Dry season (October 2020–April 2021) with 59.0% (127,213) compared to 41.0% (85,176) in the wet/rainy season (May 2021–September 2021). The total number of patients exposed to COVID-19 testing was 63.0% (2,105,002) in the rainy season and 37.0% (1,162,464) in the dry season. The dry and rainy seasons had a COVID-19 prevalence of 0.063 and 0.041 respectively. 

### 3.2. Prevalence of COVID-19 According to the Geographical Zones in Nigeria

The highest confirmed cases of COVID-19 according to the geographical zones in Nigeria were seen in the South-West, 47.5% (101,266). It also had an increased percentage of the total no of patients exposed to the COVID19 testing with 49.0% (1,539,479). The South-West was followed by 16.6% (35,465) confirmed cases in the South-East with coincidental patient testing of also 16.6% (521,537) of the total number of tested patients. South-South geographical zone in Nigeria accounted for 12.9% (27,488) of confirmed cases while testing 263,911 (9.0%), North-West with confirmed cases of 19,943 (9.3%) from 10% (314,179) of the total tested population. The North-Central had 8.8% (18,930) of the total number of confirmed cases from testing 282,761 (9.0%) patients. The lowest was the North-East geographical zone with 9267 (4.3%) cases of COVID19. The Figure 2 was obtained from testing 157,089 (5.0%) in that Zone. The prevalence of COVID19 showed 0.07% in the North-Central, 0.04% in both the North-East and North-West, 0.03% in the South-West, 0.09% in the South-South, and the highest prevalence of 0.16% in the South-East (Table 2).

### 3.3. Fatality Rate of COVID-19 during the Dry and Wet Seasons in Nigeria

The Table 3 below revealed the case Fatality rate of COVID-19 during the Dry and Wet Seasons in Nigeria. The highest fatality rate occurred during the dry season with about 1.46% from 1869 (0.160%) deaths and 127,213 cases. However, study demonstrates that in the rainy season between May-September, there was a fatality rate of 1.029% from 877 (0.041%) deaths and 85,146 confirmed COVID-19 Cases. 

### 3.4. Fatality Rate of COVID-19 during the Dry and Wet Seasons across the Geographical Regions in Nigeria

Considering the case Fatality rate of COVID-19 across geographical regions during the Dry and Wet Seasons, Table 4 below showed that the highest fatality rate seen during the dry season was seen in the North-Central region of Nigeria. Study demonstrates that the North-Central had a death toll of 196 (10.4%) out of 9457 confirmed COVID-19 cases. The highest number of deaths recorded was seen in the South-West Zone, accounting for 53.9% (1009) of the total deaths recorded from 67,501 cases during the dry season. This zone had a fatality rate of 1.49%. 246 (13.1%) deaths were documented in the South-South from total confirmed cases of 16,490, hence a fatality rate of 1.49% also. The South-East accounted to a fatality rate of 1.25% with 241 (12.8%) deaths recorded from 19,201 confirmed cases. The lowest fatality rate during the dry season was shown in the North-West with 1.12% from 8940 reported cases. This zone had 101 (5.4%) deaths. However, the lowest number of deaths documented by the study was shown in the North-East with 4.0% (76) from 5624 confirmed cases.

During the Rainy season, death toll prevailed highest in the South-West with 316 (36.0%) from 33,765 confirmed cases in the region. It had a fatality rate of 0.93%. The North-Central recorded a death toll of 52 (5.9%) from 9473 confirmed cases and a fatality rate of 0.54%, North East had 42 (4.7%) deaths out of a confirmed 3643 cases and a fatality rate of 1.15%. In the North-West, 17.6% (155) lost their lives to COVID-19 from 11,003 confirmed cases in the region. It had a fatality rate of 1.40%. South east contributed to 200 (22.8%) deaths from 16,264 confirmed cases and a rate of 1.22%. Lastly, from the South-South geographical region 112 (12.7%) were documented from 10,998 confirmed cases having a/fatality rate of 1.01% (Table 4)

### 3.5. Case Fatality (Number of Deaths) of COVID-19 and Socio Demographical Characteristics among Geographical Regions in Nigeria

Revealed in the Table 5 below, COVID-19 case fatality (Number of deaths) was highest among the males in the South-West with 1009 (76.1%) deaths, and just 23.8% (316) females, South-South had 299 (83.5%) male deaths and 59 (16.4%) were females. 62.0% (154) of deaths in the North-Central were males, while 37.9% (94) were females. North-West accounted for 170 (66.4%) male deaths and 86 (33.5%) female fatalities. In the South-East, 70.2% (310) of the deaths belonged to the males, while 29.7% (131) were female. Out of a total of 118 deaths in the North-East, 40.6% (48) were females and 59.3% (70) males. On age group, fatality of Infants-Toddlers contributed to 8 (3.2%) in the North-Central, 2 (1.6%) in the North-East, 6 (2.3%) in the North-West, none in the South-East, 5 (0.30%) in the South-West and just 1 (0.27%) in the South-South. School Age Children had 10 (4.0%) in the North-Central, 6 (5.0%) in the North-East, 22 (8.8%) in the North-West, 2 (0.4%) in the South-East, 10 (0.75%) in the South-West and 3 (0.83%) in the South-South. For Adolescents, study showed there was 22 (8.8%) in the North-Central, a single case (0.84%) in the North-East, 10 (4.0%) in the North-West, 3 (0.6%) in the South-East, a high figure to the usual in 89 (6.7%) in the South-West and 19 (5.3%) in the South-South. Death of Aged adults across the six geographical zones was distributed as follows; the North-Central had 129 (52.0%), North East 99 (83.8%), North West 137 (53.5%), South East 398 (90.2%), South West 624 (47.0%), and South-South 289 (80.7%). Considering Travel history, 94 (37.9%) in the North-Central said “Yes” while 154 (62.0%) replied “No”, 48 (40.6%) in the North East said “Yes” and 70 (59.3%) denied. In the North-west, 86 (33.5%) accepted they had a travel history while 170 (66.4%) said they did not. In the South-East, 131 (29.7%) confirmed, while 310 (70.2%) said “No”, 316 (23.8%) of the South-westerners said “Yes” while 1009 (76.1%) replied “No”. While in the South-South, 59 (16.4%) affirmed they had a travel history, 299 (83.5%) denied.

### 3.6. Socio Demographic Characteristics against Morbidity Due to COVID-19 in Nigeria

From Table 6 below, The socio demographic characteristics against morbidity due to COVID-19 illustrates the frequency was highest among the males who suffered from Respiratory distress; 992 (74.8%) and 25.1% (333) females, respondents who were sick as a result of Bronchitis were310 (70.2%) males and 131 (29.7%) females. 74.0% (265) due to Pneumonia were males while 25.9% (93) were females. Common Cold accounted for 170 (66.4%) male illnesses and 86 (33.5%) in the women. 154 (62.0%) of men were sick due to Organ dysfunction and about 94 (37.9%) women. For the total number of patients that were diagnosed with Neurological disorder, 59.3% (70) were males, however, 48 (40.6%) were revealed to be women. On age group, Infants-Toddlers diagnosed with Pneumonia had the highest percentage 8 (3.2%), 2 (0.39%) and 2 (0.15%) each for Bronchitis and Respiratory Distress. 5 (1.13%) for Cold and none for Organ dysfunction. Neurological disorder affected just a single infant. School age children within 5–10 years of age, 22 (8.8%) were unhealthy due to pneumonia, 2 (0.39%), Bronchitis, 8 (0.60%) as a result of respiratory distress, 8 (1.82%) due to cold, 2 (1.6%) and 1 (0.98%) were affected by organ dysfunction and Neurological disorder accordingly. Sickness in aged adults were shown as follows; 115 (46.3%) affected by Pneumonia, 399 (77.9%) Bronchitis, 797 (60.1%) being the highest value was demonstrated in aged adults above age 60 due to respiratory distress, Cold had contributed to the highest percentage morbidity with 384 (87.0%), 54 (45.7%) due to organ dysfunction and 46 (45.0%) were reportedly neurologically disordered. 

Considering Travel history, 94 (37.9%) affected by Pneumonia said “No” while 154 (62.0%) replied “Yes”, 48 (40.6%) down with bronchitis replied “No" and 70 (59.3%) accepted they had a travel history. On the state of illness due to Respiratory distress, 86 (33.5%) denied having a travel history while 170 (66.4%) did not. For respondents affected by cold, 131 (29.7%) denied, while 310 (70.2%) said “Yes”, 316 (23.8%) of respondents who had organ dysfunction said “No” while 1009 (76.1%) replied “Yes”. While regarding neurological disorders as morbidity cumulated (16.4%) without a travel history, and 299 (83.5%) who affirmed.

### 3.7. Morbidity Rate versus the Case Fatality Rate of COVID-19 in Nigeria

Illustrated from the Table 7 below, 358 (13.02%) deaths occurred from pneumonia due to COVID-19, Bronchitis deaths recorded were 441 (16.05%), 48.25% (1325) deaths were attributed to patients who experienced respiratory distress, Common Cold; 256 (9.32%) patients, 248 (9.03%) of the patients suffered organ dysfunction however, Neurological Disorder due to COVID-19 was responsible for 118 (4.29%) deaths. 

## 4. Discussion

Based on the findings of this study on the prevalence of COVID-19 during the dry and wet seasons in Nigeria, the study revealed that the prevalence of COVID-19 in the dry season in Nigeria (October 2020–April 2021) was 0.063. The drying and vaporisation duration of airbornes is an indicator for the illness value constant, which is determined by relative moisture and temperature both inside and outside, according to Leffler et al. [25], and the number of confirmed cases is directly related to this illness value constant. According to the findings of this study on the prevalence of COVID-19 in Nigeria during the dry season, relative moisture and temperature have an impact on coronavirus transmission in Nigeria [6]. The study revealed that the prevalence of COVID-19 infection in the wet season was 0.041, with cases of COVID-19 showing variation. This finding goes in line with meteorological studies predicting COVID-19 seasonal variation [14,15,26]. Based on the findings of this study on the seasonal prevalence of COVID-19 infection, a similar publication posited that the relations between the daily average relative humidity and temperature and the daily counted cases of COVID-19 revealed that the COVID-19 daily case counts were adversely associated with both the average relative humidity and temperature. The publication suggested that both daily relative humidity and temperature affected the incidence of COVID-19 in some districts [17]. Considering the findings of this study on the prevalence of COVD-19 in several geographical zones in Nigeria, it was demonstrated that the highest number of confirmed cases of COVD-19 was seen in the south-west with 47.5%. This could be due to the fact that the first confirmed case of COVID-19 infection was noted in a state in southwest Nigeria; thus, the area is prone to the spread of the viral infection [3]. Also, a notable increase in testing facilities by the government of Lagos State and the federal government of Nigeria was seen. The geographical prevalence of COVID-19 was 0.07% in the North Central region. This is consistent with a study by Zhu et al. [10] that found a similar prevalence of coronavirus infection in a district in Hebei. The study revealed a 0.04% prevalence of COVID-19 in both the north-east and north-west of Nigeria. This variation in prevalence could be attributed to limited testing facilities in several geopolitical zones of the country and misconceptions about the diagnosis among the general public. According to the study, 0.03% were in the south-west, which had the highest number of confirmed COVID-19 cases. The study showed 0.09% prevalence in the South-South, and the highest prevalence of COVID-19 was 0.16% in the South-East. According to findings from a few studies on face mask usage, people in the Southeast use fewer face masks, which could be attributed to rising prevalence [5,27,28]. According to a study conducted in various districts of Saudi Arabia, districts with a lower use of face masks had a higher prevalence of COVID-19 infection [8].

The mortality rate of COVID-19 is depicted as the proportion of people who die from COVID-19 among all those diagnosed with COVID-19 infection [2,26,29]. Considering the case fatality rate of COVID-19 during the dry and wet seasons in Nigeria, the findings of this study revealed that the highest fatality rate seen during the dry season was seen in the north-central region of Nigeria. The study revealed that North-Central had a death toll of 196 (10.4%) out of 9457 confirmed COVID-19 cases, hence a fatality rate of 2.07. In a recent study aimed at evaluating the effect of hot weather in hotter geographical regions, Lowen and Steel [30] discovered that the COVID-19 fatality rate is highest in dry regions. This, however, corroborates the findings of this study conducted in north central Nigeria during the dry season. The highest number of deaths recorded was seen in the South-West Zone, accounting for 53.9% of the total deaths recorded from 67,501 cases during the dry season. This zone had a fatality rate of 1.49%. South-western Nigeria includes states like Lagos, Ogun (where the first case of COVID-19 was recorded), Osun, Ekiti, and Ondo state. South-South Nigeria had a fatality rate of 1.49%. The South-East accounted for a fatality rate of 1.25%, with 241 (12.8%) deaths recorded from 19,201 confirmed cases. The North-West had the lowest fatality rate during the dry season, with 1.12% of 8940 reported cases. This zone had 101 (5.4%) deaths. However, the study found that the North-East had the fewest deaths, with a 4.0% case fatality rate. Some recent studies in line with the vitiating fatality rate of COVID-19 infection have opined that certain viruses causing respiratory infections such as coronaviruses or rhinoviruses (RV) do so in the seasons in which they are most active, especially in the spring and autumn; this season corroborates with dry season variation in Nigeria [8,9,31].

In addition to the findings of this study, the death toll was highest in the South-West during the rainy season, with a fatality rate that was 0.93% lower than during the dry season. This establishes that COVID-19 fatalities rise in the dry season compared to the wet season [19,23]. Also, the study revealed that while North Central recorded a death toll of 52 from 9473 confirmed cases and a fatality rate of 0.54% less than the 2.1% case fatality rate in the dry season, North East had a fatality rate of 1.15%. Less than the dry season’s fatality rate, and also less than the North-West’s fatality rate of 1.40%. The southeast contributed to a fatality rate of 1.22% in the wet season compared to 1.25 in the dry season. Finally, the South-South geographical region experienced a fatality rate of 1.01%, compared to 1.49 in the dry season. This shows that changes in weather are one of the factors predisposing the spread of infectious diseases [1]. Since the outbreak of the coronavirus disease 2019 (COVID-19) in Wuhan, Hubei Province, China, almost every region of the world has been battling the scourge, and its spread has yet to abate. According to the findings of this study, Nigeria has more fatalities during the dry season than during the wet season.

From this study, the sociodemographic characteristics of COVID-19-related morbidity show that the number of males who had trouble breathing was the highest (992, or 74.8%). This is in contrast with a finding by Bu et al. [19] on how gender could affect the COVID-19 infectious process. Furthermore, the study found that 74.0% (265) of pneumonia deaths were male, while 25.9% (93) were female. The common cold accounted for 170 (66.4%) male illnesses and 86 (33.5%) female illnesses. The study showed that 62.0% of the men were sick due to organ dysfunction, as were about 37.3% of the women. These findings are consistent with publications by the CDC report on demographic relationships and COVID-19 infection [2]. Considering the relationship between climatic indices and the spread of COVID-19, however, a recent publication by Zhu et al. [10] established that absolute moisture existed. It has been noted that atmospheric water content and temperature are important factors in viral spread. Dry and cold air, like influenza viruses, increases the vitality duration of those viruses, whether in droplets or on surfaces, increasing the potential for transmission [7,11]. This shows that in Nigeria, relative humidity and temperature influence the spread of the novel infection. According to the study’s findings, COVID-19 case fatality (number of deaths) was highest among males in the South-West, with 1009 (76.1%) deaths. This is consistent with a study conducted by Bogoch et al. [13] on the number of deaths associated with COVID-19 among gender disparities. According to the study, the death rate of aged adults was distributed as follows across the six geographical zones: the North-Central had 129 (52.0%), the North East had 99 (83.8%), the North West had 137 (53.5%), the South East had 398 (90.2%), the South West had 624 (47.0%), and the South-South had 289 (80.7%). This confirms previous findings that COVID-19 fatalities are associated with older adults [9,15,26]. From the study, it could be opined that 358 (13.02%) deaths occurred from pneumonia due to COVID-19; this could be due to the severity of the infection in the area. Notably, pneumonia is a common morbidity associated with most respiratory illnesses [1]. According to the study, 48.25% (1355) of the deaths were attributed to patients who experienced respiratory distress. This is consistent with a World Health Report on the morbidities associated with COVID-19 fatalities.

## 5. Conclusions

Nigeria is also a country largely affected by the global coronavirus 2019 (COVID-19) pandemic caused by coronavirus 2 (SARS-CoV-2). The study showed that the dry and rainy seasons in Nigeria had a COVID-19 prevalence of 0.063 and 0.041, respectively. Coronavirus spread is negatively correlated with temperature and absolute humidity. Nigeria, with its meteorological and seasonal variations, could be an interesting element influencing the spread and mortality rate caused by COVID-19; however, the continuous transmission and sharp rise in cases through variable humidity, from dry and cold to temperate areas, proves that meteorological changes, via an increase in moisture and temperature during dry and rainy seasons, alone would not lead to a diminution in the confirmed infection numbers without the need to intervene in the population’s healthcare in an extensive way. The uncertain overall impact of meteorological attributes on the transmission of the coronavirus demands further investigation, including how temperature and absolute moisture impact this propagation. Based on the results of this study, Nigeria has a higher mortality rate in the dry season than in the wet season. For the upcoming dry season months, the use of modelling approaches to ensure other climatic conditions analysis while taking the previously mentioned parameters into account and enveloping larger regions of the world over longer periods of time, covering several humidity and temperature ranges, could provide more accurate data and allow estimation of vulnerable areas for significant COVID-19 community transmission, and thus more targeted public health efforts. The conventional knowledge that a warmer climate might slow the spread of COVID-19 has proven inconsistent with the transmission currently evident in tropical regions of the world. We still need to stick to rigorous social enforcement measures, such as social distancing, non-medical masks to prevent droplets from entering the respiratory tract, containment, and early case detection and screening, all of which remain critical to limiting such a pandemic.

## 6. Recommendations

Based on the findings of this study, the following are recommended:i.Researchers should look at more data and meteorological analysis of how changes in the weather affect the spread of the SARS-CoV-2 virus.ii.The government should fund research and policy mechanisms that would empower health agencies to engage in advanced investigations.iii.It is extremely essential to ensure the proper implementation of careful and strict social measures, such as social distancing, continued wearing of non-medical masks to prevent the penetration of droplets into the airways, quarantine of affected and suspected patients, and early detection; this is very mandatory to promote the fight against infection and control its spread.

## Figures and Tables

**Figure 1 vaccines-11-00298-f001:**
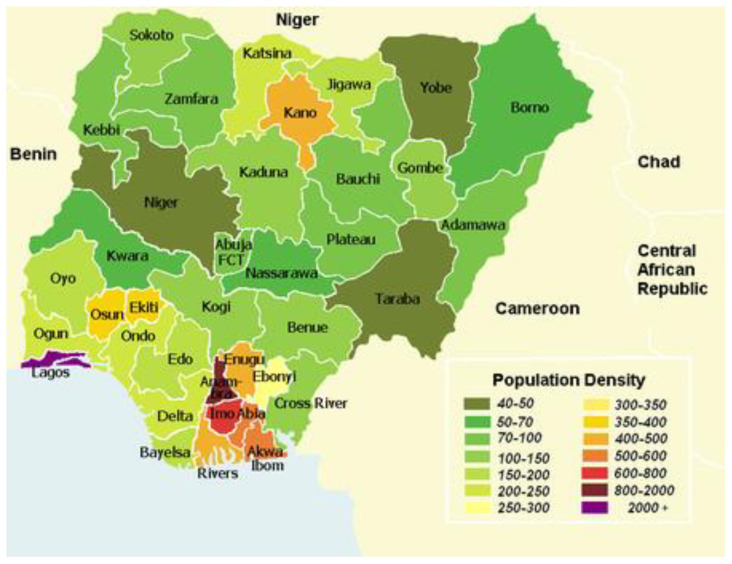
Map of Nigeria.

**Figure 2 vaccines-11-00298-f002:**
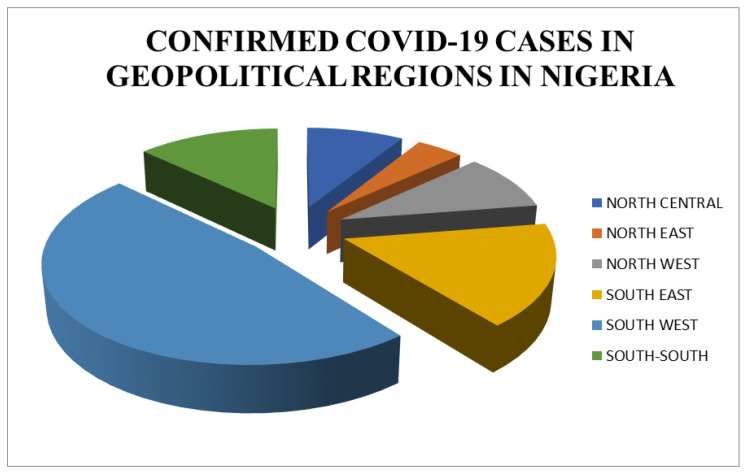
Confirmed Cases of COVID-19 in Geopolitical regions of Nigeria.

**Table 1 vaccines-11-00298-t001:** Prevalence of COVID-19 during the Dry and Wet Seasons across Nigeria.

Seasons	No of Confirmed COVID-19 Cases	Total No. Exposed to COVID-19 Testing	Prevalence of COVID-19
Dry season (October–April)	127,213 (59.0%)	1,162,464 (37.0%)	0.06
Wet (Rainy) Season (May–September)	85,176 (41.0%)	2,105,002 (63.0%)	0.041

Source: Goggle Mobility Software (2021).

**Table 2 vaccines-11-00298-t002:** Prevalence of COVID-19 according to the Geographical Zones in Nigeria.

Geographical Zones	No of Confirmed COVID-19 Cases	Total No. Exposed to COVID-19 Testing	Prevalence of COVID-19
North Central	18,930 (8.8%)	282,761 (9.0%)	0.07
North East	9267 (4.3%)	157,089 (5.0%)	0.04
North West	19,943 (9.3%)	314,179 (10.0%)	0.04
South East	35,465 (16.6%)	521,537 (16.6%)	0.16
South West	101,266 (47.5%)	1,539,479 (49.0%)	0.03
South-South	27,488 (12.9%)	263,911 (8.4%)	0.09

Source: Goggle Mobility Software (2021).

**Table 3 vaccines-11-00298-t003:** Fatality Rate of COVID-19 during the Dry and Wet Seasons in Nigeria.

Seasons	No of Deaths Due to COVID-19 Cases	No of Confirmed COVID-19 Cases	Fatality Rate
Dry season (October–April)	1869 (0.160%)	127,213	1.469%
Wet (Rainy) Season (May–September)	877 (0.041%)	85,146	1.029%

**Table 4 vaccines-11-00298-t004:** Fatality Rate of COVID-19 during the Dry and Wet Seasons across the Geographical Regions in Nigeria.

Geographical Zones	Dry Season			Wet (Rainy Season)		
	No of Deaths Due to COVID-19 Cases	No of Confirmed COVID-19 Cases	Fatality Rate of COVID-19	No of Deaths Due to COVID-19 Cases	No of Confirmed COVID-19 Cases	Fatality Rate of COVID-19
North Central	196 (10.4%)	9457	2.07%	52 (5.9 %)	9473	0.54%
North East	76 (4.0%)	5624	1.35%	42 (4.7%)	3643	1.15%
North West	101 (5.4%)	8940	1.12%	155 (17.6%)	11,003	1.40%
South East	241 (12.8%)	19201	1.25%	200 (22.8%)	16,264	1.22%
South West	1009 (53.9%)	67,501	1.49%	316 (36.0%)	33,765	0.93%
South-South	246 (13.1%)	16,490	1.49%	112 (12.7%)	10,998	1.01%

Source: Goggle Mobility Software (2021).

**Table 5 vaccines-11-00298-t005:** Case Fatality (Number of deaths) of COVID-19 and Socio demographic Characteristics among geographical regions in Nigeria.

Characteristics	Geographical Regions (Case Fatality-Number of Deaths)					
	North Central	North East	North West	South East	South West	South-South
**Sex**						
Male	154 (62.0%)	70 (59.3%)	170 (66.4%)	310 (70.2%)	1009 (76.1%)	299 (83.5%)
Female	94 (37.9%)	48 (40.6%)	86 (33.5%)	131 (29.7%)	316 (23.8%)	59 (16.4%)
**Total**	248	118	256	441	1325	358
**Age Group**						
Infants-Toddlers (0–4 yrs)	8 (3.2%)	2 (1.6%)	6 (2.3%)	0 (0.0%)	5 (0.30%)	1 (0.27%)
School Age Children (5–10 yrs)	10 (4.0%)	6 (5.0%)	22 (8.8%)	2 (0.4%)	10 (0.75%)	3 (0.83%)
Adolescent (12–17 yrs)	22 (8.8%)	1 (0.84%)	10 (4.0%)	3 (0.6%)	89 (6.7%)	19 (5.3%)
Adult (18–60 yrs)	79 (31.8%)	10 (84.7%)	81 (31.6%)	40 (9.0%)	597 (45.0%)	46 (12.8%)
Aged Adults (above 60yrs)	129 (52.0%)	99 (83.8%)	137 (53.5%)	398 (90.2%)	624 (47.0%)	289 (80.7%)
**Total**	248	118	256	441	1325	358
**Travel History**						
Yes	94 (37.9%)	48 (40.6%)	86 (33.5%)	131 (29.7%)	316 (23.8%)	59 (16.4%)
No	154 (62.0%)	70 (59.3%)	170 (66.4%)	310 (70.2%)	1009 (76.1%)	299 (83.5%)
**Total**	248	118	256	441	1325	358

**Table 6 vaccines-11-00298-t006:** Socio demographic Characteristics against Morbidity Due to COVID-19 in Nigeria.

Characteristics	Morbidities Due to COVID-19					
	Pneumonia	Bronchitis	Respiratory Distress	Cold	Organ Dysfunction	Neurological Disorder
**Sex**						
Male	265 (74.0%)	310 (70.2%)	992 (74.8%)	170 (66.4%)	154 (62.0%)	70 (59.3%)
Female	(25.9%)	131 (29.7%)	333 (25.1%)	86 (33.5%)	94 (37.9%)	48 (40.6%)
**Total**	358	441	1325	256	248	118
**Age Group**						
Infants-Toddlers (0–4 yrs)	8 (3.2%)	2 (0.39%)	2 (0.15%)	5 (1.13%)	0 (0.0%)	1 (0.98%)
School Age Children (5–10 yrs)	22 (8.8%)	2 (0.39%)	8 (0.60%)	8 (1.82%)	2 (1.6%)	1 (0.98%)
Adolescent (12–17 yrs)	24 (9.6%)	17 (3.32%)	29 (2.18%)	3 (0.6%)	15 (12.7%)	19 (18.6%)
Adult (1860 yrs)	79 (31.8%)	92 (17.9%)	489 (36.90%)	40 (9.0%)	47 (39.8%)	35 (34.3%)
Aged Adults (above 60 yrs)	115 (46.3%)	399 (77.9%)	797 (60.1%)	384 (87.0%)	54 (45.7%)	46 (45.0%)
**Total**	248	512	1325	441	118	102
**Travel History**						
Yes	154 (62.0%)	70 (59.3%)	170 (66.4%)	310 (70.2%)	1009 (76.1%)	299 (83.5%)
No	94 (37.9%)	48 (40.6%)	86 (33.5%)	131 (29.7%)	316 (23.8%)	59 (16.4%)
	248	118	256	441	1325	358

**Table 7 vaccines-11-00298-t007:** Morbidity Rate versus the Case Fatality Rate of COVID-19 in Nigeria.

Morbidities	Case Fatality Rate in Nigeria
Pneumonia	358 (13.02%)
Bronchitis	441 (16.05%)
Respiratory Distress	1325 (48.25%)
Cold	256 (9.32%)
Organ Dysfunction	248 (9.03%)
Neurological Disorder	118 (4.29%)

## Data Availability

Data can be made available on reasonable request by first or corresponding author.

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
