# Peer review of "Seasonal Variation and Geographical Distribution of COVID-19 across Nigeria (March 2020–July 2021)"

_vaccines, 2023, doi:10.3390/vaccines11020298_

Round 1
Reviewer 1 Report
The paper is interesting and well written. The paper evaluate the geographical distribution of COVID-19 in Nigeria. I suggest to discuss if different food administration may impact the different incidence (I.E., vitamin D, microbioma?; see and add as references papers by Murdaca et concerning vitamin D, microbioma and COVID 19). Second, a briefly description of vaccination in Nigeria (see and add as reference paper by Murdaca et al concerning vaccination in autoimmune diseases; individuals in Nigeria may be at risk of lowering immune responses for limited food administration, stress for different style of life etc).
Specific Comments:
1. This study aimed to determine the seasonal variation and geographical distribution of Covid’19 across Nigeria.
2 The topic original is relevant in the field
3. The study adds concluded that prevalence of covid-19 varies in seasons in 48 Nigeria
4. No specific improvement needs, No further controls are needed
5. The conclusions are consistent with the evidence and arguments presented and they address the main question posed
6. The references are appropriate
7. Nothing in particular comments on the tables and figures.
Author Response
The paper is interesting and well written. The paper evaluates the geographical distribution of COVID-19 in Nigeria. I suggest to discuss if different food administration may impact the different incidence (I.E., vitamin D, microbioma?; see and add as references papers by Murdaca et concerning vitamin D, microbioma and COVID 19). Second, a briefly description of vaccination in Nigeria (see and add as reference paper by Murdaca et al concerning vaccination in autoimmune diseases; individuals in Nigeria may be at risk of lowering immune responses for limited food administration, stress for different style of life etc).
Author’s Response: Thank you very much for your input. The Reference paper was added accordingly as directed by the reviewer and all structural changes were made.
Reviewer 2 Report
The work entitled:" SEASONAL VARIATION AND GEOGRAPHICAL DISTRIBUTION OF COVID-19 ACROSS NIGERIA (MARCH, 2020- 3JULY, 2021)" done by Moroh etal is not organized and does not presented well although the topic is of great importance however, there are major points should be addressed:
1- in the introduction, the research question and the research hypothesis are need to be clarified; no well information are written
2- In introduction, the model explaination of the the relation between sensitive airbornes with relative humidity and temperature is not clear.
3- In introduction, the author need to clarify and summarize the previous similar publications.
4- The aim of work is broad and not concise.
5- Methods is messy and the readers will be confused when reading it,, need to be more organized.
6- English errors, please go over the manuscript.
Author Response
#Reviewer 2
In the introduction, the research question and the research hypothesis are need to be clarified; no well information are written
Ans: changes have been made in the manuscript as per your suggestions
In introduction, the model explaination of the the relation between sensitive airbornes with relative humidity and temperature is not clear.
Ans: changes have been made in the manuscript as per your suggestions
In introduction, the author need to clarify and summarize the previous similar publications.
Ans: changes have been made in the manuscript as per your suggestions
The aim of work is broad and not concise.
Ans: changes have been made in the manuscript as per your suggestions and the aim of the work is clear
Methods is messy and the readers will be confused when reading it,, need to be more organized.
Ans: changes have been made in the methodology as per your suggestions
English errors, please go over the manuscript.
Ans: changes have been made in the manuscript as per your suggestions
Reviewer 3 Report
Section A: Minor Comments
The paper needs more efforts to be more suitable for potential publication. More typos were discovered through the process of reading the manuscript. Some of them are mentioned below:
1. Line 32, "Corona Virus Infection" should be "Corona virus infection".
2. Lines 35, 36, 44, 299 and 347, "Fatality" should be "fatality".
3. Lines 38, 48 and 313, "the Dry season" should be "the dry season".
4. Lines 42 and 44, "the North-Central" should be "the North-central".
5. Lines 42 and 238, "the North-Central" should be "the North-central".
6. Line 49, modify the sentence "Nigeria Hence; further Data and Meteorological ...." to be "Nigeria. Hence, further data and meteorological ....".
7. Line 50, "Cov-2 Virus spread" should be "Cov-2 virus spread".
8. Line 71, "Middle East respiratory syndrome" should be "middle East respiratory syndrome".
9. Line 127, "Xie& Zhu, 2020" should be "Xie & Zhu, 2020".
10. Line 146, "Sajadiet al., 2021" should be "Sajadiet et al., 2021".
11. Rewrite the lines from 163 to 172. The presentation is not good.
12. Rewrite the lines from 186 to 199. The presentation is not good.
13. Why lines 201 and 202 are bold?.
14. Line 246, "March to September, ..." NOT bold.
15. Line 535, (Paezet al., 2020; Auleret al., 2020, ....) should be "(Paezet et al., 2020; Auleret et al., 2020, ....)".
16. Line 538, WHO??. The abbreviation should be added before using it.
Section B: Statistics Comments
1. Non-parametric plots should be sketched and discussed in detail. These figures help the analysts to read the behavior of data.
2. Did the authors test the normality of data?. Discuss
3. Did the authors test the homogeneity of data? Discuss.
4. Why didn't the authors utilize the regression approach as a support tool?.
Section C: Other Comments
1. Instrument and Method Data Collection Section should be revised and modified.
2. More details about Table 4 should be listed.
3. Abbreviation Section should be listed.
4. References should be revised.
Author Response
#Reviewer 3
The paper needs more efforts to be more suitable for potential publication. More typos were discovered through the process of reading the manuscript. Some of them are mentioned below:
Ans: changes have been made in the manuscript as per your suggestions
Line 32, "Corona Virus Infection" should be "Corona virus infection".
Ans: changes have been made in the manuscript as per your suggestions
Lines 35, 36, 44, 299 and 347, "Fatality" should be "fatality".
Ans: changes have been made in the manuscript as per your suggestions
Lines 38, 48 and 313, "the Dry season" should be "the dry season".
Ans: changes have been made in the manuscript as per your suggestions
Lines 42 and 44, "the North-Central" should be "the North-central".
Ans: changes have been made in the manuscript as per your suggestions
Lines 42 and 238, "the North-Central" should be "the North-central".
Ans: changes have been made in the manuscript as per your suggestions
Line 49, modify the sentence "Nigeria Hence; further Data and Meteorological ...." to be "Nigeria. Hence, further data and meteorological ...."
Ans: changes have been made in the manuscript as per your suggestions
Line 50, "Cov-2 Virus spread" should be "Cov-2 virus spread".
Ans: changes have been made in the manuscript as per your suggestions
Line 71, "Middle East respiratory syndrome" should be "middle East respiratory syndrome".
Ans: changes have been made in the manuscript as per your suggestions
Line 127, "Xie& Zhu, 2020" should be "Xie& Zhu, 2020".
Ans: changes have been made in the manuscript as per your suggestions
Line 146, "Sajadiet al., 2021" should be "Sajadiet et al., 2021".
Ans: changes have been made in the manuscript as per your suggestions
Rewrite the lines from 163 to 172. The presentation is not good.
Ans: changes have been made in the manuscript as per your suggestions
Rewrite the lines from 186 to 199. The presentation is not good.
Ans: changes have been made in the manuscript as per your suggestions
. Why lines 201 and 202 are bold?.
Ans: bold letters have been removed and changes have been made in the manuscript as per your suggestions
Line 246, "March to September, ..." NOT bold.
Ans: changes have been made in the manuscript as per your suggestions
Line 535, (Paezet al., 2020; Auleret al., 2020, ....) should be "(Paezet et al., 2020; Auleret et al., 2020, ....)".
Ans: changes have been made in the manuscript as per your suggestions
Line 538, WHO??. The abbreviation should be added before using it.
Ans: changes have been made in the manuscript as per your suggestions
Section B: Statistics Comments
- Non-parametric plots should be sketched and discussed in detail. These figures help the analysts to read the behavior of data.
Giving to the nature of the statistical data obtained from secondary sources, the researcher have been made methodological comments regarding the analysis method in the manuscript as per your suggestions
- Did the authors test the normality of data?. Discuss
It was not discussed and changes have been made in the manuscript as per your suggestions
- Did the authors test the homogeneity of data? Discuss.
Data presented was from varying sources and a clear descriptive method was used in the study.
- Why didn't the authors utilize the regression approach as a support tool?.
A clear descriptive method was used in obtaining and reporting the findings and changes were noted in the manuscript as per your suggestions
Section C: Other Comments
- Instrument and Method Data Collection Section should be revised and modified.
Ans: changes have been made in the manuscript as per your suggestions
- More details about Table 4 should be listed.
Ans: changes have been made in the manuscript as per your suggestions
- Abbreviation Section should be listed.
Ans: changes have been made in the manuscript as per your suggestions
- References should be revised.
Ans: changes have been made in the manuscript as per your suggestions
We appreciate the editor’s thoughtful consideration and we have addressed all the issues mentioned by the editors and reviewers, and we have conducted any extra analyses as requested and reported them in the revised version. We are prepared to make modifications based on your advice and recommendations if there are still any concerns.
Thank you again for all your input, guidance and help which has improved the manuscript.
Yours Sincerely,
Round 2
Reviewer 2 Report
It is ok now
Reviewer 3 Report
All comments have been done.